# Pharmacists’ Role in Global TB Elimination: Practices, Pitfalls, and Potential

**DOI:** 10.3390/healthcare12111137

**Published:** 2024-06-03

**Authors:** Alina Cernasev, Jonathan Stillo, Jolie Black, Mythili Batchu, Elaina Bell, Cynthia A. Tschampl

**Affiliations:** 1Department of Clinical Pharmacy and Translational Science, College of Pharmacy, University of Tennessee Health Science Center, Nashville, TN 37211, USA; acernase@uthsc.edu; 2Department of Anthropology, Wayne State University, 656 W. Kirby St. 3054 FA/B, Detroit, MI 48202, USA; jonathan.stillo@wayne.edu (J.S.); hp4356@wayne.edu (E.B.); 3Schneider Institutes for Health Policy and Research, The Heller School for Social Policy and Management, Brandeis University, Waltham, MA 02453, USA; jolieblack@brandeis.edu; 4Department of Public Health, Dr. Kiran C. Patel College of Osteopathic Medicine, Nova Southeastern University, Davie, FL 33313, USA; mb4084@mynsu.nova.edu

**Keywords:** tuberculosis (TB), literature review, pharmacy, pharmacist, global, USA, TB elimination, pharmaceutical care

## Abstract

Tuberculosis (TB) is the top infectious killer in the world despite efforts to eliminate it. Pharmaceutical care roles are pillars of pharmacy practice, and pharmacists are well equipped to serve a unique role in the pathway to provide education about TB. Previous systematic reviews emphasize pharmacists’ role in treating TB; however, pharmacists can and do play much broader roles in overall TB elimination efforts. Five researchers searched five electronic databases (PubMed, PsychInfo, CINAHL, Academic Search Premier, and Embase). Search terms included pharmacy, pharmacist, tuberculosis, antitubercular agents, supply, distribution, and drug therapy. Inclusion criteria were studies published from 2010 through March 2023, in English or Spanish, addressed a specific TB-related role for pharmacists/pharmacies, and were peer-reviewed. Exclusion criteria included pharmacology, pharmacokinetics, clinical trials on drug efficacy, and editorials. Two researchers conducted each level of review; for discordance, a third researcher reviewed, and a decision was reached by consensus. Roles were extracted and cross-referenced with traditional pharmaceutical care steps. Of the initial 682 hits, 133 were duplicates. After further review, we excluded 514 records, leaving 37 articles for full extraction. We found nine roles for pharmacists in TB prevention and classified them as implemented, not implemented, or recommended. These roles were: (1) TB symptom screening; (2) Referring to TB care systems; (3) TB testing; (4) Dispensing TB medication correctly and/or directly observed therapy; (5) Counseling; (6) Looking to reduce socioeconomic barriers; (7) Procurement of TB medications; (8) Quality assurance of TB medications; (9) Maintaining and using pharmacy data systems. Pharmacists are well situated to play a vital role in the global fight against TB. Findings suggested pharmacists in many settings have already expanded their roles related to TB elimination beyond traditional pharmaceutical care. Still others need to increase the understanding of TB procurement and treatment, their power to improve TB care, and their contributions to data systems that serve population health. Pharmacy curricula should increase TB-related training to better equip future pharmacists to contribute to TB elimination.

## 1. Introduction

Dr. Robert Koch identified tuberculosis (TB) in 1882. Despite this, TB continues to be one of humanity’s greatest killers [1]. Initially, treatment consisted of good nutrition, rest, and other non-pharmacological agents until the discovery of streptomycin and para-amino salicylic acid (PAS) in the 1940s and isoniazid in the 1950s (Ryan, 1993). By the mid-1960s, a four-drug regimen, consisting of rifampicin, isoniazid, pyrazinamide, and ethambutol was in use worldwide and is still widely used to this day [2]. The development of this four-drug regimen would significantly reduce TB rates globally; however, this progress was uneven and would be further complicated by poverty, the HIV/AIDS pandemic, and increasing rates of drug resistance that required much more complicated treatment. Nevertheless, global efforts to develop new effective TB medications and vaccines stalled following the success in lowering TB rates in wealthier countries, contributing to a more than forty-year gap between the development of rifampicin and the next novel TB drug, bedaquiline [3].

Despite the availability of treatments for even highly drug-resistant strains of TB, TB surpassed HIV/AIDS in 2015 to become the most significant single infectious killer in the world [4]. While COVID-19 temporarily surpassed TB in terms of total deaths, TB continues to claim nearly 1.5 million lives every year [5]. An estimated two billion people are infected with TB, and the Global Tuberculosis Report 2023 emphasized an increase in reported TB cases, with 10.6 million in 2022 compared to 10.3 million in 2021 and 10.1 million in 2020 [5]. This reversed the two-decade trend of slow annual decreases [6]. The situation is more alarming in some countries due to the high incidence and prevalence of TB; resulting in a list of “high-burden countries for TB” from the WHO [7]. Indeed, this dire situation has led some to question whether TB has ever been “controlled” in much of the developing world [2].

The Centers for Disease Control and Prevention (CDC) reported that 13 million people in the United States of America (USA) live with TB infection [8]. The CDC published preliminary 2023 data in March 2024 that showed a 16% increase in TB cases, due largely to consequences of the COVID-19 pandemic, where resources and attention were diverted from TB programs [9].

TB causes human misery and continues to burden the public health infrastructure. TB affects all genders, ages, and strata of society. Adult men represent 55% of all cases reported worldwide, and women and children make up the remaining 33% and 12%, respectively [10]. However, previous research has strongly suggested that the lower rates of TB seen among women is a sign of underdiagnosis and a comparative lack of access to health care [11,12]. Women, in particular, are profoundly affected by TB, especially during pregnancy and childbearing, with recent data finding TB to be a significant contributor to maternal and perinatal mortality and morbidity [13,14].

In the last decade, pharmacists around the world have broadened their scope of practice and integrated as part of interprofessional teams that include physicians, nurses, social workers, and laboratory staff. Given their unique pharmaceutical expertise, pharmacists have been recognized as important players in the healthcare system. This recognition was paved during the 1990s when the “pharmaceutical care concept” emerged and set the tone for current pharmacy practice [15]. Pharmaceutical care was defined by Hepler and Strand as “the responsible provision of drug therapy for the purpose of achieving definite outcomes that improve a patient’s quality of life” [15]. The core of pharmaceutical care is to maximize a patient’s therapeutic outcomes by utilizing pharmacists’ skills and knowledge via seven steps: (1) establishment of a pharmacist–patient relationship; (2) patient data collection, analysis, and interpretation; (3) finding out medication-related problems; (4) establishing therapy outcomes and goals together with the patient; (5) setting possible pharmacotherapy alternatives and selecting the best plan, preparing a monitoring plan; (6) implementation of the individual pharmacotherapy regimen and monitoring plan; (7) follow-up [15].

Previous systematic reviews focused on pharmacists’ role in treating TB [16]; however, there is a paucity of studies to focus on pharmacist roles in overall TB elimination efforts, particularly regarding actions beyond the core of pharmaceutical care, i.e., the interpersonal level.

## 2. Methods

This rapid scoping review was conducted to identify the existing literature and literature gaps focused on TB-related pharmacist interventions globally and in the USA. We then analyzed the results using newly developed criteria by our team to address the role of pharmacists in pharmaceutical care. A descriptive synthesis from both qualitative and quantitative studies is presented in Table 1 and Table 2.

### 2.1. Search Strategy

Five electronic databases (PubMed, PsychInfo, CINAHL, Academic Search Premier, and Embase) were searched independently and in duplicate over four months in 2023. The search strategy used the following keywords of “pharmacist or pharmacy”, “TB or tuberculosis”, “antitubercular agents”, and “supply and distribution”. To ensure the search strategy was yielding the appropriate research outcomes, a librarian was consulted throughout the process.

**Table 1 healthcare-12-01137-t001:** Summary of addressed TB-related roles for pharmacists and other study details by article.

First Author, Year	Country	WHO Classification *	World Bank Country Income Classification ^$^	Study Design	Population Characteristics	Pharmacy Practice Site Level Discussed or Implicated	Roles Played by the Pharmacist/Pharmacy in the Study ^§^	Hepler and Strand (1990) [15] Pharmacy Care Steps ^¶^
Abimbola et al., 2015 [17]	Nigeria	High burden	LMI	Quantitative	TB patients diagnosed at three Nigerian health centers.	Community	Achieved 2; Gap identified 1, 6; Recommended 2	e, f
Bate et al., 2013 [18]	19 countries	All burden levels	LMI, UMI	Quantitative	Private pharmacies in 19 low- and middle-income countries.	Community	Achieved 7; Gap identified 8; Recommended 8	No criteria match
Bell et al., 2012 [19]	Cambodia	High burden	LMI	Qualitative	Data from 54 pharmacy owners in referral program for themes and improvement suggestions.	Community	Achieved 1, 2, 5, 6	a, b, g
Bell, Duncan, et al., 2015 [20]	Cambodia	High burden	LMI	Qualitative	Assessed stakeholders’ perceptions of TB detection program.	Community	Achieved 1, 2, 5	a
Bell, Ilomäki, et al., 2015 [21]	Cambodia	High burden	LMI	Quantitative	180 Phnom Penh pharmacies interviewed face-to-face.	Community	Achieved 1, 2, 5; Recommended 5	a, b
Bell et al., 2016 [22]	Cambodia	High burden	LMI	Quantitative	Referral program involved participating pharmacy employees.	Community	Achieved 1, 2; Gap identified 4	a, b, e, g
Bradley et al., 2015 [23]	South Africa	High burden	UMI	Mixed methods	Cape Town pharmacists aided HIV/AIDS, TB, and chronic diseases.	Hospital, health clinic, community	Achieved 4, 5, 8, 9; Recommended 9	b, d, e, f
Colvin et al., 2014 [24]	Tanzania	High burden	LMI	Mixed methods	Trained healers, pharmacists referred TB cases for evaluation.	Health clinic	Achieved 1, 2	a, b
Cowan et al., 2013 [25]	Ethiopia	High burden	LI	Qualitative	Healthcare workers in five Ethiopian hospitals.	Hospital	Achieved 1, 3, 4; Gap identified 3, 7; Recommended 7	a, b, c, e, f
Ehsanul Huq et al., 2018 [26]	Bangladesh	High burden	LMI	Mixed methods	15 years and older patients with smear positive TB.	Health clinic	Gap identified 1, 2, 3, 4; Recommended 2	b, e, f
Frederick et al., 2021 [27]	India	High burden	LMI	Mixed methods	Enablers and barriers to implementation, including resources and partnerships.	Hospital, community	Achieved 1, 4, 5, 9; Gap identified 1, 4, 5, 9; Recommended 9	a, b, f
Gayathri et al., 2020 [28]	India	High burden	LMI	Qualitative	Community pharmacists in Tiruvallur district of Tamil Nadu, India.	Community	Achieved 2, 5; Gap identified 1, 3, 4, 5	a, b, f, g
Gnanasan et al., 2011 [29]	Malaysia	High burden	UMI	Mixed methods	TB patients with comorbid diabetes mellitus.	Hospital	Achieved 4, 5	b, c, d, e, f
Jakeman et al., 2015 [30]	USA	Not high burden	HI	Quantitative	New Mexico patients interested in TB testing from 2011 to 2013.	Community	Achieved 1, 2, 3	a, b, g
Jakeman, Logothetis, Saba, et al., 2020 [31]	USA	Not high burden	HI	Quantitative	Study followed LTBI patients in New Mexico.	Community	Achieved 2, 3, 9	a, b, g
Jakeman, Logothetis, Roberts, et al., 2020 [32]	USA	Not high burden	HI	Mixed methods	DOTS in community pharmacies; licensed pharmacist followed a protocol.	Community	Achieved 4, 5, 9	a, b, c, e, f, g
Karuniawati et al., 2019 [33]	Indonesia	High burden	LMI	Quantitative	75 respondents divided into three groups: counseling, counseling with leaflets, and control.	Community	Achieved 5, 9	b, c
Kwabla et al., 2022 [34]	Ghana	High burden	LMI	Mixed methods	Community medicine outlets (CMOs) operators.	Community	Achieved 1, 2; Gap identified 1, 2, 4; Recommended 2, 6	c
Lara-Júnior et al., 2022 [35]	Brazil	High burden	UMI	Quantitative	Brazilian city offered pharmacotherapeutic follow-up for TB in a public health clinic.	Health clinic	Achieved 4, 5	c, d, e, f, g
Lopes et al., 2017 [36]	Brazil	High burden	UMI	Mixed methods	62 patients followed in one outpatient clinic. 53% were females. The median age was 51 years old.	Community	Achieved 5	c, d, g
Magadzire et al., 2014 [37]	South Africa	High burden	UMI	Qualitative	Nurses, pharmacists, and doctors (36 providers from 6 sectors).	Community	Achieved 6; Recommended 4, 6, 7	c, d, f
McKennon et al., 2016 [38]	USA	Not high burden	HI	Mixed methods	University of Washington pharmacy students received TB screening module.	Community	Achieved 1, 3	a
Mhalu et al., 2019 [39]	Tanzania	High burden	LMI	Mixed methods	Patients that were lost to diagnostic follow-up (LDFU) patients.	Community	Achieved 1, 2, 9	b
Millard et al., 2018 [40]	India	High burden	LMI	Quantitative	Private retail spaces in Maharashtra state.	Community	Achieved 7	f
Múñiz-González et al., 2012 [41]	Spain	Not high burden	HI	Quantitative	Retrospective study analyzed data from 351,086 inhabitants.	Hospital; other	Achieved 9	b
Noor et al., 2021 [42]	Pakistan	High burden	LMI	Quantitative	Hospitalized TB patients from Pakistan and surrounding areas, including Afghanistan, with potential drug interactions.	Hospital	Recommended 5	c, e
Paydar et al., 2011 [43]	100 countries	All burden levels	N/A	Quantitative	Surveyed TB programs in 100 countries.	Health clinic	Achieved 4, 7; Gap identified 6; Recommended 4	f
Pradipta et al., 2021 [44]	Indonesia	High burden	LMI	Qualitative	TB patients, physicians, nurses, pharmacists, TB activist, TB programmers at the district and primary care levels.	Hospital, health clinic, community	Gap identified 4; Recommended 5	c, f
Rakesh et al., 2021 [45]	India	High burden	LMI	Mixed methods	Kerala study involved 33 stakeholder interviews.	Health clinic	Achieved 4, 9; Recommended 9	b, f
Rezende Macedo do Nascimento et al., 2017 [46]	Brazil	High burden	UMI	Mixed methods	Verified 50 items selected from the Relação Nacional de Medicamentos Essenciais (Rename—National List of Essential Medicines) of 2012 in Brazil.	Health clinic, community	Gap identified 7,8	f
Sarker et al., 2017 [47]	Bangladesh	High burden	LMI	Qualitative	Rural and urban extrapulmonary TB patients in Bangladesh.	Community	Achieved 1; Gap identified 2	No criteria match
Sintayehu et al., 2022 [48]	Ethiopia	High burden	LMI	Mixed methods	Evaluated first-line TB medications available in capital of Ethiopia.	Hospital, health clinic	Achieved 9; Gap identified 7; Recommended 9	No criteria match
Sterling et al., 2020 [49]	USA	Not high burden	HI	Qualitative	All in the US impacted by TB infection.	Hospital, health clinic, community, other	Recommended 4	e
Tang et al., 2018 [50]	China	High burden	UMI	Mixed methods	Patients with first-time pulmonary tuberculosis in China.	Community	Achieved 1, 4, 5	a, b, d, f, g
Ullah et al., 2020 [51]	Pakistan	High burden	LMI	Quantitative	Individuals with presumptive TB seeking treatment at community pharmacies in Pakistan.	Community	Achieved 1, 2, 5, 9; Gap identified 4; Recommended 2, 9	b, e, f
Wong et al., 2023 [52]	Malaysia	High burden	UMI	Mixed methods	Survey was completed by 388 community pharmacists, and 23 pharmacists participated in the interview.	Community	Achieved 2, 5; Recommended 4, 5	f
Zawahir et al., 2021 [53]	Vietnam	High burden	LMI	Quantitative	Vietnam, private pharmacies.	Hospital	Gap identified 1, 2, 4	f

* The World Health Organization (WHO) defines countries with “high burden countries for TB” as “the top 20 countries in terms of their estimated absolute number of new (incident) cases in 2019; plus the 10 countries with the most severe burden in terms of the incidence rate (new cases per 100,000 population in 2019) that are not already in the top 20”, and that meet a minimum threshold of 10,000 new cases per year for TB. Read more at: https://cdn.who.int/media/docs/default-source/hq-tuberculosis/who_globalhbcliststb_2021-2025_backgrounddocument.pdf?sfvrsn=f6b854c2_9 (accessed on 14 May 2024). ^$^ The World Bank has defined four income categories for countries: Low-income (LI), Lower-middle-income (LMI), Upper-middle-income (UMI), and High-income (HI). Read more details at https://datahelpdesk.worldbank.org/knowledgebase/articles/906519-world-bank-country-and-lending-groups (accessed on 14 May 2024). **^¶^** Hepler and Strand (1990) [15] pharmaceutical care steps are (a) establish a pharmacist–patient relationship; (b) patient data collection, analysis, and interpretation; (c) finding out medication-related problems; (d) establishing therapy outcomes and goals together with patient; (e) setting possible pharmacotherapy alternatives and selecting the best plan, prepare a monitoring plan; (f) implement individual regimen and monitoring plan (dispense); (g) follow-up. ^§^ Roles for pharmacies/pharmacists documented by the authors: (1) TB symptom screen/ID potential TB; (2) Refer people to TB care systems; (3) TB testing; (4) Dispensing TB medication correctly and/or directly observed therapy; (5) Counseling, including managing adverse reactions and side effects; (6) Looking to reduce socioeconomic barriers for people in need of TB treatment; (7) Procurement of TB medications; (8) Quality assurance of TB medications; (9) Maintaining and/or using pharmacy data systems for population health. For each role, we marked whether the study described achievements, gaps, and/or recommendations around each role.

### 2.2. Inclusion and Exclusion Criteria

After removing duplicate results, all authors screened studies for inclusion and exclusion criteria. Eligible studies were included if they generally pertained to pharmacists’ interventions in any one or more countries, particularly if the study addressed pharmacists aiding with adherence to TB regimens, managing TB side-effects and drug interactions, addressing drug shortages, an/or participating in community-based prevention efforts to combat TB. The primary focus was pharmacists’ roles and activities conducted globally in service of TB care. A secondary focus emerged where we mapped the traditional pharmaceutical care roles to the TB-related roles and activities we initially extracted.

Qualitative, quantitative, and mixed-methods studies were included if they focused on care for people undergoing TB treatment, provided a role for pharmacists or pharmacies was explicitly stated. Other inclusion criteria were: studies where training or didactic information was provided for pharmacists or pharmacy students, studies published in 2010 and later in English or Spanish, and conference abstracts.

Studies were excluded if their focus was on the pharmacology or pharmacokinetics of TB regimens; presented clinical trials for efficacy; or were news articles, commentaries, editorials, or systematic literature reviews.

Results from the initial search were validated by the research team and imported into Zotero and Excel. After initial importation, the duplicate records were removed by the entire research team. The research team assured each of the remaining abstracts was reviewed by two researchers independently to ensure the studies met the eligibility criteria. After the preliminary review of the titles and abstracts, three researchers reviewed the full text of the articles to determine their eligibility. If any disagreement occurred regarding the inclusion or exclusion of studies, the team met and discussed the article until consensus was finalized. The PRISMA flow diagram [54] illustrates this process (Figure 1).

### 2.3. Data Abstraction and Synthesis

The research team met multiple times to discuss the data extraction from the full-text articles and developed, by consensus, the criteria to include country and population studied, WHO classification of high TB burden [7], World Bank country income designation [55], study design, pharmacy practice level (community, hospital, etc.), and roles of the pharmacist/pharmacy (Table 1). The roles were listed in two ways; first, in accordance with the pharmaceutical care model [15] and second, according to our own, broader extraction criteria. Further, we indicated whether each article discussed each role as achieved, exhibiting a gap, and/or as a recommendation. If 50% or more of the pharmacies/pharmacists in a study undertook a given role, “achieved” was listed; if <50% did not, “Gap” was listed. If a study described multiple components related to a single role, it is possible to see both “Achieved” and “Gap” indicated for the same study and role. If the authors wrote any recommendation related to the role, “recommendation” was listed independent of any achievements or gaps already identified.

Three researchers independently assessed each study and agreed on the final data. Then, the full research team compiled the extracted data into a relevant classification system and conceptual model to address current needs in TB. The findings of these studies are presented in Table 1. We used non-stigmatizing language throughout the manuscript based on Stop TB Partnership’s Words Matter guide [56].

**Table 2 healthcare-12-01137-t002:** TB-related care roles for pharmacists discussed in 37 articles according to the role being in evidence, missing, and/or recommended by the authors.

	Role 1: TB Symptom Screen/ID Potential TB			Role 2: Refer People to TB Care Systems			Role 3: TB Testing			Role 4: Dispensing TB Medication Correctly and/or Directly Observed Therapy			Role 5: Counseling, including Managing Adverse Reactions and Side Effects			Role 6: Looking to Reduce Socioeconomic Barriers for People in Need of TB Treatment			Role 7: Procurement of TB Medications			Role 8: Quality Assurance of TB Medications			Role 9: Maintaining and Using Pharmacy Data Systems for Population Health		
Study	Achieved	Gap	Recommend	Achieved	Gap	Recommend	Achieved	Gap	Recommend	Achieved	Gap	Recommend	Achieved	Gap	Recommend	Achieved	Gap	Recommend	Achieved	Gap	Recommend	Achieved	Gap	Recommend	Achieved	Gap	Recommend
Abimbola et al., 2015 [17]	No	Yes	No	Yes	No	Yes										No	Yes	No									
Bate et al., 2013 [18]																			Yes	No	No	No	Yes	Yes			
Bell et al., 2012 [19]	Yes	No	No	Yes	Yes	No							Yes	No	No	Yes	No	No									
Bell, Duncan, et al., 2015 [20]	Yes	No	No	Yes	No	No							Yes	No	No												
Bell, Ilomäki, et al., 2015 [21]	Yes	No	No	Yes	No	No							Yes	No	Yes												
Bell et al., 2016 [22]	Yes	No	No	Yes	No	No				No	Yes	No															
Bradley et al., 2015 [23]										Yes	No	No	Yes	No	No							Yes	No	No	Yes	No	Yes
Colvin et al., 2014 [24]	Yes	No	No	Yes	No	No																					
Cowan et al., 2013 [25]	Yes	No	No				Yes	Yes	No	Yes	No	No							No	Yes	Yes						
Ehsanul Huq et al., 2018 [26]	No	Yes	No	No	Yes	Yes	No	Yes	No	No	Yes	No															
Frederick et al., 2021 [27]	Yes	Yes	No							Yes	Yes	No	Yes	Yes	No										Yes	Yes	Yes
Gayathri et al., 2020 [28]	No	Yes	No	Yes	No	No	No	Yes	No	No	Yes	No	Yes	Yes	No												
Gnanasan et al., 2011 [29]										Yes	No	No	Yes	No	No												
Jakeman et al., 2015 [30]	Yes	No	No	Yes	No	No	Yes	No	No																		
Jakeman, Logothetis, Saba, et al., 2020 [31]				Yes	No	No	Yes	No	No																Yes	No	No
Jakeman, Logothetis, Roberts, et al., 2020 [32]										Yes	No	No	Yes	No	No										Yes	No	No
Karuniawati et al., 2019 [33]													Yes	No	No										Yes	No	No
Kwabla et al., 2022 [34]	Yes	Yes	No	Yes	Yes	Yes				No	Yes	No				No	No	Yes									
Lara-Júnior et al., 2022 [35]										Yes	No	No	Yes	No	No												
Lopes et al., 2017 [36]													Yes	No	No												
Magadzire et al., 2014 [37]										No	No	Yes				Yes	No	Yes	No	No	Yes						
McKennon et al., 2016 [38]	Yes	No	No				Yes	No	No																		
Mhalu et al., 2019 [39]	Yes	No	No	Yes	No	No																			Yes	No	No
Millard et al., 2018 [40]																			Yes	No	No						
Múñiz-González et al., 2012 [41]																									Yes	No	No
Noor et al., 2021 [42]													No	No	Yes												
Paydar et al., 2011 [43]										Yes	No	Yes				No	Yes	No	Yes	No	No						
Pradipta et al., 2021 [44]										No	Yes	No	No	No	Yes												
Rakesh et al., 2021 [45]										Yes	No	No													Yes	No	Yes
Rezende Macedo do Nascimento et al., 2017 [46]																			No	Yes	No	No	Yes	No			
Sarker et al., 2017 [47]	Yes	No	No	No	Yes	No																					
Sintayehu et al., 2022 [48]																			No	Yes	No				Yes	No	Yes
Sterling et al., 2020 [49]										No	No	Yes															
Tang et al., 2018 [50]	Yes	No	No							Yes	No	No	Yes	No	No												
Ullah et al., 2020 [51]	Yes	No	No	Yes	No	Yes				No	Yes	No	Yes	No	No										Yes	No	Yes
Wong et al., 2023 [52]				Yes	No	No				No	No	Yes	Yes	No	Yes												
Zawahir et al., 2021 [53]	No	Yes	No	No	Yes	No				No	Yes	No															

Notes: If 50% or more of the pharmacies/pharmacists in a study undertook a given role, “Achieved” was marked yes. If <50% did not, “Gap” was marked as yes. If a study described multiple components related to a single role, it is possible to see both “Achieved” and “Gap” marked yes. If the authors wrote any recommendation related to the role, “Recommend” was marked yes independently of the other two columns. Blank cells indicate that the authors did not describe or discuss the corresponding role.

## 3. Results

### 3.1. Results Using the Pharmacist’s Roles

Through this rapid scoping review, we identified nine roles discussed for pharmacists or pharmacies. Findings are presented according to each of the nine roles, including details related to achievements, gaps, and recommendations.

#### 3.1.1. Role 1: TB Symptom Screen/ID Potential TB

Out of 37 studies included in this review, 14 studies described a successful role played by the pharmacist in screening and identifying potential TB cases.

Several studies showed how pharmacists act as frontline healthcare workers, making them instrumental in identifying potential TB cases [24,25,34,38,39,47,50,51]. In a series of studies on Cambodia’s pharmacy referral program led by Bell et al., pharmacists performed and achieved various pharmaceutical roles including TB screening [19,20,21,22]. A study that tested pharmacist behavior provided information on how private pharmacies operating under Cambodia’s TB Referral Program would address actors, known as “standardized patients”, coming into pharmacies with symptoms suggestive of TB [22]. The study found nearly all respondents referred the “standardized patients” to the TB clinic, in accordance with Cambodia’s national guidelines. This study demonstrated the effectiveness of Cambodian pharmacists in TB screening for referral. Additionally, Frederick et al. demonstrated how the vital role played by pharmacists in screening led to the creation of a pharmacy-based surveillance program and notification system in South India [27]. In the USA state of New Mexico, the integrative role of community pharmacists was demonstrated as pharmacists screened and identified potential TB cases [30].

#### 3.1.2. Role 2: Refer People to TB Care Systems

Twelve of the 37 studies in this review described successfully referring people to TB care systems.

A 2012 study conducted by Bell et al., evaluated the attitudes and practices of pharmacy owners offering pharmacy-initiated referral services in Phnom Penh from 2005 to 2010 following Cambodia’s implementation of a public/private mix (PPM) intervention program [19]. The authors found that altruism, rather than financial gain, played a central role in owners’ decisions to participate and continue in the referral program. Ongoing professional support, improved public sector care, and government media campaigns were cited by owners as ways to improve their ability to refer people living with TB [19]. Bell et al. further assessed the perceptions of organizational stakeholders on program collaboration/participation and sustainability [20]. They found that stakeholders thought the referral program made an impact in improving TB case detection rates due to its accessibility given the large, pre-existing network of private pharmacies in Cambodia, as well as strong government commitment, collaborative relationships, and management structures. The authors agreed the referral program’s sustainability may rely on moving away from dependence on external agencies [20]. Another study by Bell et al. examined the factors associated with pharmacy providers’ decision to refer or not refer clients [21]. The study reported that provider commitment and years in the referral program were correlated with increased referral rates, but the presence of infection control measures was not.

Bell and colleagues continued their research by presenting a hypothetical scenario of a client with symptoms suggestive of TB based on training received through Referral Program Workshops [22]. The survey found that 92% of the study’s 180 participants would have referred the hypothetical client to a public sector clinic [22]. Similarly, Colvin et al. worked on a community-based project in Tanzania aimed at improving TB case notification [24]. As in Cambodia’s referral program, pharmacists and traditional healers were given training on recognizing symptoms of TB and referring symptomatic individuals to diagnostic facilities. From 2009 to 2011, this referral network contributed 38% to 70% of new TB case notifications, and 97% of individuals referred went to diagnostic facilities [24]. Aiming to increase TB case detection in Pakistan, Ullah et al. created a formalized referral pathway for use in community pharmacies in Gujurat, Lahore, and Sheikhupura Districts [51]. The study found that 85% of the 500 participating pharmacies remained active in referring clients during this period and 9% of all new TB cases registered in the districts originated from the referrals of these participating pharmacies [51].

In 2011, New Mexican pharmacists who completed the training through the New Mexico Department of Health were granted the authority to prescribe, administer, and read tuberculin skin tests (TSTs). Jakeman et al. found most recipients of the TST had been referred by pharmacy personnel [32]. Abimbola et al. in Nigeria found the number of people transferred to appropriate TB care providers was highest when people with TB symptoms contacted pharmacy providers versus other non-formal health services indicating a relatively strong referral link [17]. Mhalu et al. in Tanzania reported referrals to TB clinics from pharmacies as a matter of course [39].

Barriers to referral were emphasized in five studies. A qualitative study by Sarker et al. found that most people were referred to direct observation of therapy services (DOTS) clinics by government facilities and private centers [47]. In a cross-sectional study in Bangladesh examining care pathways of people being treated for TB, Ehsanul Huq et al. found that people who sought advice from non-qualified providers (45.9% of which visited pharmacies), only 18.2% were referred to the health complex where they later received treatment [26]. In Vietnam, Zawahir et al. evaluated the rate of referral at private pharmacies using “standardized patients” presenting a person with presumptive TB or MDR-TB [53]. While pharmacists who had asked about the standardized patients’ medical histories were significantly more likely to refer to providers, the study found that, overall, only 12.3% of standardized patients with presumptive TB and 36.3% of standardized patients with presumptive MDR-TB were referred for further medical assessment [53]. Bell et al. found that pharmacy owners believed a lack of commitment from public sector clinics and community perceptions of pharmacies as “drug suppliers only” had negative impacts on the credibility of their referrals [19]. A study in Ghana by Kwabla et al. reported that major barriers to referral included fear of negative responses by referred people, personal negative attitudes, lack of training on TB, and a lack of financial and logistical support [34].

Four studies offered additional recommendations for improving pharmacy referral. Ullah et al. suggested setting up certain pharmacies as DOTS providers to increase sector involvement in TB care and encourage referral. Additionally, the regulation of anti-TB drugs and increased incentives for pharmacies were proposed as a way to facilitate referrals in Pakistan [51]. In contrast, Kwabla et al. found that permitting the sale of some over-the-counter TB medicines promoted a wider range of dispensing options that would advance the number of referrals [34]. Ehsanul Huq et al. suggested that all medical practitioners be integrated into the National Tuberculosis Control Program to improve the overall referral system [26]. Other recommendations to increase rates of referral included increasing the education of pharmacy staff on the disease and the importance of obtaining an accurate medication and health history [53].

#### 3.1.3. Role 3: TB Testing

Four of the 37 studies in this review described pharmacists successfully taking on the role of TB testing.

All three studies that aimed to directly expand pharmacists’ roles to include TB testing were successful [30,31,38]. From 2011 to 2013, 578 people received TB tests from New Mexico pharmacists with a reported follow-up rate of 92.8% [30]. Community pharmacies were found to be an ideal setting for testing services given accessibility, pharmacy operating hours, and lack of required appointments [30]. Between 2014 and 2016, 1709 people received TB testing at 12 New Mexican pharmacies. Of the people who participated in the study, 98.2% reported having a positive experience, and 91.1% stated they would return to the pharmacy for future TB tests [31]. In 2014, the University of Washington School of Pharmacy implemented a two-week training course on TB education, screening, TST administration, and test assessment for second year pharmacy students. The course significantly improved students’ knowledge of TB and willingness to perform TB testing [38].

Three other studies discussed the role of pharmacists in TB testing indirectly. In India, Gayathri et al. found that only 33.6% of pharmacists had sufficient knowledge of TB testing [28]. In Ethiopian hospitals, surveyed healthcare providers were found to possess satisfactory knowledge of TB diagnostics [25]. In Bangladesh, Ehsanul Huq et al. found that 58.9% of TB people living with TB had received drugs but no diagnostic testing from their initial provider, who included pharmacists [26].

#### 3.1.4. Role 4: Dispensing TB Medication Correctly and/or Directly Observed Therapy

Nineteen of the 37 articles discussed the correct dispensing of TB medications and/or implementing DOTS at pharmacies.

In New Mexico, a pilot study used community pharmacists to treat people with latent TB infections using combination weekly therapy with isoniazid and rifapentine [32]. It demonstrated these medications can be safely administered via pharmacy with a high rate of treatment completion [32]. Lara-Junior et al. assessed the implementation and effectiveness of pharmacotherapeutic follow-up services on TB treatment and adherence in Brazil [35]. Through these services, pharmacists provide consultations, conduct follow-ups, prescribe over-the-counter medications, and request exams for people undergoing TB treatment. Similar benefits were found by Tang et al. in their study evaluating the effectiveness of pharmaceutical care among people undergoing TB treatment in China [50]. Participating pharmacists were responsible for ensuring appropriate medication dosages and prescriptions. People who received pharmaceutical care were found to have greater treatment success [50].

Surveillance programs in India were shown to influence the dispensing of TB medications. One study found that the sale of anti-TB drugs to people without definitive TB diagnosis had decreased since the implementation of increased education on TB drug control [45]. In another study examining the effects of India’s surveillance policies, Frederick et al. revealed that data collected from pharmacy reports showed that 1307 and 1673 people had purchased anti-tuberculosis drugs at pharmacies in Dharmapuri and Salem districts, respectively [27]. Action research conducted by Gnanasan et al. showed pharmacists also played an important role in the care and treatment of people with concurrent TB and diabetes mellitus in Malaysia [29].

A survey of national TB drug control policies found that 96 of 100 responding countries had some form of DOTS implemented, and 44 allowed private pharmacies to dispense TB drugs [43]. In Ethiopia, Cowan et al. found that hospital staff were proud of the success rates of their respective outpatient DOTS programs, crediting these accomplishments to program decentralization from tertiary to regional hospitals [25]. Data submitted from Pakistani pharmacies and follow-ups suggested the benefit of pharmacy staff as DOTS facilitators for people undergoing TB treatment, especially in cases of relapsed TB [51].

Bradley et al. evaluated participatory action research conducted by the authors in Cape Town, South Africa from 2008 to 2011 [23]. The study focused on the roles and competencies of district and sub-district pharmacists during transitionary periods. One of the main roles of both district and sub-district pharmacists was found to be the coordination and monitoring of pharmaceuticals. Moreover, sub-district pharmacists assisted in dispensing medicines at clinics due to a lack of employee coverage. This was found to be problematic by research participants since it limited sub-district pharmacists’ ability to engage in management activities.

Several studies showed a gap in pharmacists’ ability to dispense TB medications correctly. In Vietnam, Zawahir et al. found that 92.3% of pharmacies sold antibiotics to standardized patients with presumptive TB and 96.9% sold antibiotics to standardized patients with presumptive MDR-TB [53]. Similarly, Kwabla et al. found that 43.6% of surveyed pharmacists and over-the-counter medicine sellers in Ghana agreed that colleagues chose to not refer symptomatic customers due to a desire to profit from the sale of their own medicines [34]. Although the Indian government provides free TB medications, Gayathri et al. found that 51% of people undergoing TB treatment still went to community pharmacies for TB medications; of these instances, 34% of pharmacists dispensed upon request [28]. In Indonesia, stigma, particularly as perceived coming from health providers such as pharmacits was found to be a barrier to successful TB treatment [44]. Furthermore, 24% of participants trained through Cambodia’s referral program would still have sold an antibiotic at the time of referral [22]. Paydar et al. suggested that uncontrolled access to rifampicin has contributed to the prevalence of MDR-TB [43].

With regard to recommendations, a mixed methods study aimed at exploring the feasibility of DOTS programs in community pharmacies in Malaysia found pharmacists agreed the availability of DOTS at community pharmacies would increase the effectiveness and adherence of TB treatment [52]. Additionally, US-based guidelines for treating TB infection identified three different rifamycin-based regimens as preferred options for treating people with TB infection [49]. The guidelines emphasized that rifampicin and rifapentine are not interchangeable and urged prescribers and pharmacists to ensure correct medication selection [49]. Paydar et al. also recommended the WHO encourage countries to implement restrictions on the private sector sale of TB medications to help combat the emergence of MDR-TB [43].

#### 3.1.5. Role 5: Counseling, including Managing Adverse Reactions and Side Effects

Fourteen of the 37 studies in this review described pharmacists successfully performing counseling services related to TB education, testing, prevention, and/or medications.

In Cambodia, Bell et al. found pharmacy owners viewed counseling people with presumed TB and motivating them to accept referral as important parts of their job, despite fears of contracting TB during consultations [19]. Bell and colleagues also found the majority of participating pharmacy staff reported they had counseled clients with symptoms of TB [21] and that organizational stakeholders of Cambodia’s referral program strongly supported the program [20]. Pharmacists participating in a study in India provided health education to people with presumed TB 45.7% of the time and believed that doing so was an important part of their service to the community [28]. In Malaysia, 78.4% of surveyed pharmacists believed pharmacists could assist with monitoring for TB drug side effects, and 75.3% felt that pharmacists could provide medication counseling to people undergoing TB treatment [52].

Studies in this review overwhelmingly demonstrated the benefits of pharmacist counseling. Gnanasan et al. found that pharmacists in a Malaysian hospital often acted as “communication vehicles” between physicians and people living with TB and diabetes mellitus [29]. Similarly, Jakeman et al. found that New Mexican pharmacists were able to assist in the management of individuals’ adverse drug events via communication and collaboration with the New Mexico Department of Health [32]. With this counseling, 71% of participants who experienced adverse drug effects still completed treatment [32]. The effects of pharmacotherapeutic follow-up and other pharmacist interventions on drug-related problems (DRPs) were also examined by Lopes et al. in a study in Brazil [36]. They demonstrated pharmacists were able to solve the majority of DRPs through guidance on drug usages and risks [36]. Another study in Brazil found that pharmacotherapeutic follow-up services significantly increased cure rates of people living with TB [35].

In a quantitative study among TB patients at a hospital in Indonesia, Karuniawati et al. found that treatment adherence increased from 44% to 84% over a 6-month period when both pharmacist counseling and informational leaflets were provided to patients [33]. Tang et al. reported higher rates of successful treatment and patient adherence when Shanghai patients were provided with patient education and counseling on medications [50]. In Pakistan, Ullah et al. looked at shifting conventional models of pharmaceutical care to encompass a “counseling and triage” role [51]. During the process of identification and referral, pharmacists were asked to complete referral slips for clients consenting in the study. Clients were provided with a copy of this document to present to referred clinics. If clients did not visit the referred clinic, pharmacy staff provided counseling during subsequent pharmacy visits, resulting in a client referral uptake of 63% [51].

The role of pharmacists in counseling was only opposed by one study, and this was regarding surveillance programs. In India, it was found that repeated counseling was necessary to convince patients to provide pharmacies with identifying information required by India’s TB elimination surveillance program [27]. However, participants were not in support of this practice and felt prescribing physicians should either obtain this information or provide this counseling themselves, rather than pharmacists [27]. Another study highlighted a transition away from direct patient care by district and sub-district pharmacists in favor of more managerial positions during structural reorganization in South Africa [23].

Four studies offered recommendations relevant to pharmacists’ role in counseling. Since adverse drug reactions were found to be a major barrier to TB treatment in Indonesia, Pradipta et al. suggested the use of pharmacists as drug consultants to improve treatment success through direct patient involvement and education on adverse drugs reactions [44]. Noor et al. also recommend clinical pharmacist participation in hospitals to manage drug–drug interactions in TB patients via the evaluation of patient profiles, medications, and screening for potential drug–drug interactions [42]. Wong et al. strongly recommended in-depth counseling be provided to TB patients enrolling in video DOT services to ensure appointment adherence [52], and Bell et al. recommended the implementation of a 10-min, purpose-designed consultation protocol in Cambodian pharmacies [21].

#### 3.1.6. Role 6: Looking to Reduce Socioeconomic Barriers for People in Need of TB Treatment

Five of 37 studies described barriers or facilitators to help reduce socioeconomic barriers that people with TB face.

In Cambodia, people with presumptive TB often had a fear of referrals due to distance and travel costs [19]. Pharmacy owners saw their role as shepherding presumptive clients from diagnosis to treatment and cure [19]. Similarly, socio-economic barriers affecting access to medicines in South Africa revolved around the cost, safety, and availability of transportation [37]. To address these barriers, “stable patients” were often prescribed medicines for longer durations with instructions to check in if they experienced medical issues before their next appointment. Facility managers and pharmacy personnel supported this flexible approach to prescribing policies [37]. Another way to address the socio-economic barriers faced by TB patients was advocated by Kwabla et al. after analyzing feedback from Ghanian operators of community medication outlets [34]. The authors recommended financial enablers for people needing TB treatment [34]. Abimbola et al. pointed to unnecessary costs incurred by inappropriate consultations before the TB diagnosis and recommended decentralizing TB care into the community level [17].

#### 3.1.7. Role 7: Procurement of TB Medications

Three articles discussed achieving successful outcomes in the procurement of TB medications, three identified challenges, and two offered additional recommendations.

Sintayehu et al. looked at the factors affecting TB medication shortages in Ethiopia via interviews of pharmacy facility managers and department heads [48]. The study found that 62.7% of facilities were out of stock of at least one first-line TB medication, most commonly rifampicin 75 mg/isoniazid 50 mg tablets [48]. TB medication shortages were found to be attributable to delayed delivery and lowered stock at supplying pharmaceutical warehouses. The authors recommended the Pharmaceuticals Supply Agency (EPSA) and other partners “work collaboratively to improve their service” and ensure heath facilities are able to access a sufficient supply of TB medications [48].

Supply chain issues were also seen in South Africa, where delivery of medications to local clinics was a substantial challenge [37]. Similarly, in a cross-sectional survey of pharmaceutical services in the Brazilian Unified Health System, it was found that inadequate financial resources, delays, disorganization, and problems in the pharmaceutical market were the main causes of frequent medication shortages [46]. Over one-third (38%) of medicine dispensers in the Brazilian Unified Health System reported medication shortages took place always or repeatedly [46]. Inadequate infrastructure was found to correlate with decreased medication availability [37].

Drug procurement practices were linked to MDR-TB by two studies in this review [18,43]. Paydar et al. found a slight majority of lower to middle income countries with high rates of MDR-TB were found to lack adequate control policies for TB medications [43]. In a study on private pharmacies in 17 countries, Bate et al. found that pharmacy procurement and dispensation of substandard medicines may result in the development of additional drug resistance [18].

Millard et al. reported that rifampicin was available at only 64.5% of locations in India [40]. The authors recommended that India’s regulatory body review the number of brands on the market and develop a centralized database of registered pharmaceutical products. While Ethiopia does have a national procurement list, Cowan et al. noted at the time of their study that pediatric dosages were not on this list and highlighted the need for improved procurement practices and availabilities of these dosages [25].

#### 3.1.8. Role 8: Quality Assurance of TB Medications

Three of the 37 studies explicitly discussed the quality of the medications for potential dispensing.

In a study assessing the quality of isoniazid and rifampicin in private pharmacies in 17 countries, Bate et al. reported that of the 713 samples they collected, 9.1% failed basic quality testing for the level of active pharmaceutical ingredient (API) or disintegration. The failure rate was 16.6% among 11 African countries, 10.1% in India, and 3.9% in the other middle-income countries [18]. They noted that the presence of falsified and low-quality drugs likely contributes to antibiotic resistance and recommended large-scale studies in all markets [18]. In South Africa, Bradley et al. identified district and sub-district level pharmacists’ participation in quality assurance activities and noted that in previous immunization campaigns, stock had been wasted due to expiration [23]. Finally, a study from Brazil found gaps in the supply chain management of essential medicines [46].

#### 3.1.9. Role 9: Maintaining and Using Pharmacy Data Systems for Population Health

Ten studies depicted the usefulness of pharmacy data as a key input for improved management of cases involving TB, for TB-related research, and/or for assisting a larger disease surveillance system, all of which are inputs toward the health of the broader population.

Jakeman et al. tracked adverse drug events experienced by patients receiving TB treatment in New Mexico pharmacies and found patients who finished their TB treatment were less likely to have experienced an adverse drug event than those who discontinued treatment [32]. In Tanzania, Mhalu et al. used pharmacy data to examine delays in care seeking and loss to diagnostic follow-up (LDFU) [39]. Sintayehu et al. used pharmaceutical stock data to examine the determinants of first line TB drug stock-outs in Ethiopia [48]. Pharmacy data were also utilized by Múniz-González et al. to compare the TB incidence seen in Spain’s regional epidemiological surveillance system register (SIVE) to that suggested by prescription data of isoniazid and rifampicin fixed-dose combination pills. In both 2008 and 2009, the regional incidence rates were far lower than the examined data suggested [41]. Data were generated by pharmacists implementing patient questionnaires in a study by Karuniawati et al. who reviewed the effectiveness of educational leaflets and counseling on TB treatment adherence [33].

Two studies noted the limitations of India’s new pharmacy surveillance system. Participants of Frederick et al. in Tamil Nadu observed difficulties in distinguishing between prescriptions for individual pills versus fixed-dose combinations [27]. Additionally, patients were often reluctant to share personal information making it challenging to determine why they were prescribed TB medications and thereby confusing the reported number of patients being treated for TB versus for other conditions [27]. Rakesh et al. argued that prescribing information for H1 list drugs (which can only be sold by a licensed pharmacist to an individual with a prescription) can be used to identify “missing” cases of TB in India, such as those where individuals are treated by private providers [45]. Additionally, they recommended education efforts targeting the providers dispensing these medications and increased use of pharmaceutical data to inform policy and improve TB standards of care [45].

The relevance of pharmacists’ role in surveillance was also examined by Bradley et al. in their study on the activities of district and sub-district level pharmacists in South Africa [23]. These were found to include the coordination and monitoring of pharmaceuticals, quality assurance, and clinical governance, as well as research for district-level pharmacists and medication dispensing for sub-district level pharmacists. In New Mexico, Jakeman et al. found that having community pharmacists perform and read TB tests can serve to provide improved data as well as patient referrals that might otherwise be lost [31]. Ullah et al. also argued that community pharmacies are a useful resource for TB case detection and referral, especially in places such as Pakistan which have large but fragmented private sectors [51].

## 4. Discussion

This scoping review aimed to provide an in-depth analysis of all pharmacists’ roles related to TB elimination as documented in the literature, including those outside the scope of the traditional pharmaceutical care model. We extracted items from 37 studies and identified nine roles that pertain to interpersonal, organizational, community, and national relationships. Furthermore, the studies were conducted in both high- and low-TB burdened countries, focused on pharmacies in all setting types, and included countries from all but the lowest income level.

One major finding is the preponderance of studies that provided evidence of the critical roles played by pharmacists across the globe in identifying and referring possible TB cases. Pharmacists in various countries, such as Cambodia, South India, and the USA, have been successful in screening and identifying potential tuberculosis cases, playing a vital role as frontline healthcare workers [19,20,22,24,25,27,30,34,38,39,47,50,51]. However, pharmacy referral rates varied greatly [22,24,26,47,51,52,53] suggesting the presence of significant barriers in certain countries and regions and the need for improved knowledge of and training on the stages of TB among pharmacy staff [28,34,53]. This was supported by high referral and case identification rates in studies where referral programs were supplemented with pharmacy staff training on how to recognize and refer individuals with symptoms of TB [21,24,51]. A lack of financial and/or logistical support for pharmacies providing referral services was also mentioned in several studies as a limitation to increasing the number of referred TB cases [17,34,51]. Pharmacists surveyed in multiple countries expressed support for referral programs involving pharmacies [19,21,34,52], even to the extent of seeing it as a social responsibility to “contribute to preventing spread of diseases in their communities” [34]. Not surprisingly, in a study conducted in India, barriers to identification and referral included the volume of people with presumed or confirmed TB and pharmacy workload [28].

A few of the studies that focused on the critical role of pharmacists in the referral system were qualitative, presenting more evidence of how pharmacists play a crucial role in linking people to the TB cascade of care [19,21,22]. Furthermore, these studies provided in-depth details on how the regional areas might benefit from improved infrastructure and reduced TB cases.

Testing for TB and treating TB infection (LTBI) are natural next steps after pharmacists can successfully screen and refer potential TB cases based on symptomatology. A Collaborative Practice Agreement (CPA) developed in New Mexico, USA allowed pharmacists to broaden their scope of practice to include the prescription of TB testing in community pharmacies [31]. The authors found that pharmacies were ideal settings for TB testing services due to their accessibility, and pharmacists were able to successfully perform and read TB tests with high rates of approval and follow-up from people receiving care [31]. Furthermore, the CPA expanded the scope of practice to allow pharmacists in New Mexico to administer TB preventative therapies using weekly doses of isoniazid and rifapentine (the “3HP” regimen), which resulted in completion rates comparable to traditional administration [32]. This type of scope of practice expansion is particularly useful in places like the USA where public health infrastructure, including specialized public health employees, has been defunded and deteriorated for decades [57].

Pharmacies in several countries regularly dispensed drugs, such as antibiotics, to people with presumptive TB prior to any testing or diagnostics [22,26,28,53]. However, two studies highlighted how pharmacies can be ideal settings for the pharmaceutical care of people with TB when drug dispensation is accompanied with medication counseling and supported by additional pharmacist education on TB [32,35]. Education is especially important given how inappropriate sales of medicines like quinolone antibiotics, cough suppressants, and steroids can mask TB symptoms [53,58,59] and given the new TB regimens being developed [6,60].

In 2022, the World Health Organization approved a shorter, four-month regimen for DS-TB that consists of moxifloxacin, rifapentine, isoniazid, and pyrazinamide [6]. This regimen represents the first improvement to the standard, four-drug, six-month regimen for DS-TB since the 1960s. However, there are challenges to its adoption, such as ensuring the accessibility of rifapentine, for which there is greater demand than ever [60]. Another potential challenge is that moxifloxacin, up to this point, has been reserved for treating DR-TB. Therefore, clinicians who typically treat DS-TB, especially in places where DR-TB rates are low may have little to no experience using fluoroquinolones, particularly a fourth generation one like moxifloxacin to treat TB. Another potential issue in adopting this regimen is the introduction of rifapentine into a treatment regimen as, even though it is in the same family as rifampicin, the two drugs are not interchangeable and cannot be switched when the prescription is filled and dispensed [49]. Thus, introducing these new guidelines presents several potential challenges which will require health care teams to ensure the appropriate medication regimen is prescribed and dispensed to the person undergoing TB treatment.

Pharmacists’ role in DOTS was mentioned in two studies [32,52]. While the use of DOTS is not appropriate for everyone with TB [61], the WHO guidelines state that, when utilized, DOTS should be conducted at a person’s preferred location and by a professional of their choosing [62]. Allowing community pharmacists to conduct DOTS may be one approach to achieve these guidelines, thereby avoiding some of the substantial time burdens and human right threats that DOTS has the potential to create. This is supported by the results of Jakeman et al., which reported improved outcomes when using community pharmacists in DOTS [32].

Fourteen studies highlighted the successful role of pharmacists in providing counseling services related to tuberculosis (TB) education, testing, prevention, and/or medications, showing that pharmacists can effectively educate people receiving TB care, motivate them to accept referrals, and improve treatment adherence through counseling [19,20,21,23,27,28,29,32,33,35,36,42,44,50,51,52]. These studies included recommendations to enhance pharmacist involvement in counseling to address barriers to TB treatment and improve individuals’ outcomes. A particular concentration of studies discussed the importance of counseling around adverse drug reactions [29,32,33,35,36,42,50]. Here, improvement is needed as well, as others have called for better counseling on side effects, both before medicines are dispensed and when adverse events occur [63].

Although less frequent, we were pleased to find strategies that pharmacists took to help reduce socioeconomic barriers facing people undergoing TB treatment. They acknowledged that travel distances to treatment centers and associated costs were found to be of key concern to people with TB [19,37], and recommended that public health officials increase the number of TB treatment sites to increase accessibility and reduce travel costs [34]. It is, therefore, of note that pharmacists’ involvement in the healthcare team has been shown to decrease total health expenditures, unnecessary care, and societal costs [64].

Procurement and quality assessment of TB medicines shifts the focus from interpersonal interactions to local, subnational, and national system interactions. Five studies examining the procurement of TB drugs found that pharmacies commonly experienced inadequate availability of essential medicines, often due to supply chain issues [25,37,40,46,48]. This is highly problematic given the connection of TB medication procurement to a person’s ability to access prescribed treatment regimens, which can ultimately lead to increased risk of developing drug-resistant TB. Recommendations included collaborative efforts and regulatory review to combat drug resistance [25,37].

Quality assurance of TB medications was also found to be poor in African countries and India compared to other middle-income countries, which can further contribute to higher rates of drug resistance [18]. Additionally, the studies reviewed showed linkages between the prevalence of MDR-TB and unregulated access to anti-TB medications in private pharmacies [43], as well as suggested delays in effective TB treatment when pharmacists provide other, non-prescribed medications to symptomatic clients [26]. Three studies focused on ensuring medication quality, with findings revealing a notable failure rate in basic quality testing, particularly in African countries [18,23], India [18], and other middle-income countries [18,46], further highlighting concerns regarding antibiotic resistance and the need for large-scale studies. Nevertheless, in South Africa, district and sub-district pharmacists were involved in quality assurance activities to prevent stock wastage [23]. Although life-saving TB medications are available, one may argue that additional rigorous regulations regarding medications that do not meet quality standards must be implemented to save more lives. It is also worth noting the limited number of studies focused on children’s formulations and the availability of these TB regimens to the pediatric population. Given pharmacists’ specialized expertise in medication formulation, further studies could include this aspect to address concerns with limited access to these populations.

Finally, ten studies highlighted the valuable role of pharmacy data in managing care, TB-related research, and TB surveillance systems, showcasing its utility in tracking adverse drug events, examining delays in care seeking, stock-outs of TB drugs, and contributing to population health, among other benefits [23,27,31,32,33,39,41,45,48,51]. These studies also used pharmacy-generated data to compare TB incidence rates across sub-national regions, identify limitations of surveillance data, and recommend improvements to pharmacy surveillance systems for better data accuracy and policy development [23,27,45,51]. A recent editorial built on this idea by emphasizing the need to capitalize on digital advancements [65]. We found several articles in the initial stage of our review that only used pharmacy data to calculate disease burden. While they did not meet inclusion criteria, they reinforced the findings from Múñiz-González et al. in underscoring the importance of pharmacy data, and the utility of quality improvement in this area [41].

One task we did not find mentioned in the literature either in Role 4 (dispensing/DOT) or Role 8 (procurement), was related to assuring access to newer TB medicines. The advances in TB regimens for both DS and DR-TB offer potential benefits to many people who would have had to endure longer and more toxic treatments, but these scientific advances will only improve health if they are available to the people who need them. The right to enjoy the benefits of scientific advancement (the right to science) requires that tangible goods such as medicines and vaccines (not just intangible benefits) must be made available to all people [66]. This right is highly regarded to the extent that some have argued that without achieving the right to science, achieving the right to health is not possible [67]. Moreover, strategies found in this review that aimed at removing barriers to TB care and bringing it closer to people are often only time or geographically limited projects. In comparison, operational research is rarely scaled up to the state or national levels, suggesting areas for further research of these barriers.

In the last three decades, Hepler and Strand’s understanding of pharmaceutical care has been embraced by pharmacists globally [68,69]. For example, in the USA, the federal Omnibus Budget Reconciliation Act of 1990 (OBRA-90) addressed federally funded and state-managed Medicaid programs, including pharmacy services (The Omnibus Budget Reconciliation Act of 1990, P. L. 101–508, 104 Stat. 1388, Sec. 4401 OR Pub L No. 101–508, §4401, 1927(g) November 5, 1990; OBRA 1990 Regulations, Federal Register). The OBRA-90 Act, which took effect in 1993, required pharmacists to provide counseling to all people during the prescription point of sale. Subsequently, the responsibility of counseling individuals became part of the American Pharmacist Association Standard of Practice for the Profession of Pharmacy [70]. Additionally, pharmacists in the USA must now have in place a protocol called a Collaborative Practice Agreement (CPA) with a prescriber to initiate changes to the patients’ medication regimen(s). The CPAs vary by state, and the pharmacists’ responsibilities differ significantly [71,72]; however, we found evidence this CPA mechanism was used to increase TB elimination efforts in the USA.

At the present time in the USA, TB testing is not generally in the scope of practice for pharmacists. Nevertheless, the results from Jakeman et al. showed a very practical example of how pharmacists in the USA can increase their contributions to the goal of TB elimination by expanding beyond the Helpler and Strand model [30,32]. Future studies could explore additional partnerships and legislation to maximize pharmacists’ scope of practice. However, this may not be an appropriate next step for all countries.

Since pharmacy approaches and TB burdens differ around the globe, the priority activities that pharmacists contribute to the elimination of TB need varying supportive policies as well. For example, the scope of practice in Nigeria focuses mainly on dispensing and counseling, and pharmacists are not licensed to provide additional clinical services such as diagnostic tests [17].

While the scope of pharmacy practice varies across countries it is increasingly integrated into team-based care, where pharmacists are routinely involved in providing vaccination services, smoking cessation counseling, adherence to antiretroviral treatment, and screening for disease states [73,74]. They are well situated to contribute to the TB cascade of care by addressing the customers’ needs and questions about treatment [34,39]. Due to their extensive medication and clinical knowledge, pharmacists can work collaboratively with other healthcare providers to assess care plans for people receiving health care and to suggest appropriate changes for improved outcomes [31,39].

## 5. Limitations and Strengths

One of the limitations of this review was the exclusion of “grey literature.” This might have resulted in the exclusion of projects where pharmacies partnered with governments or non-governmental organizations, but that were published on websites or in the gray, rather than peer-reviewed literature and therefore missed by this review. Additionally, certain studies might have been missed because they might not have been correctly indexed in the databases, resulting in selection bias. Various studies included in this review did not present the research paradigm or theoretical framework used and might not have defined the pharmaceutical care role described by Hepler and Strand. Thus, we may have had difficulty categorizing those articles using the Pharmaceutical Care roles. A final limitation is that only English and Spanish language studies were included. This may have resulted in missing studies from high TB burden places that were published in other languages. We also note many of the articles we found spoke about important factors for pharmacists to know in terms of pharmacotherapy, co-morbidities, etc. While critical to understand and important for the overall discussion toward TB elimination, those items fell outside the scope of our research question and were not included in the final analysis.

The strengths of this scoping review are the inclusion of studies published in both English and Spanish, a comprehensive search of five databases across three University library systems, and the development of an innovative criteria for data synthesis that was uniquely adapted for TB. Additionally, we had at least two researchers conduct every level of review (e.g., title, abstract, full) and did not limit data extraction to the traditional interpersonal level. Furthermore, we found evidence from all pharmacy settings, both high and low TB-burdened countries, and three of four World Bank income categories. In sum, we provide actionable, precedented ideas for pharmacists and pharmacy leaders in different countries and clinical contexts that will allow them to make even bigger impacts on the health of their communities.

## 6. Conclusions

Elimination of TB globally is urgent, and pharmacists are well positioned to play important roles in the fight against this air-borne disease. This review has identified best practices already occurring in some places, as well as areas which can be improved to better care for people with TB in various settings.

Future practice guidelines should provide clear and concise guidance on how to care for people who present with a cough, should be backed by local pharmacy associations, and include structured mechanisms to support people who may have TB [65]. They should also enhance pharmacists’ awareness about medication shortages and increase their scope of practice in ways that are most appropriate given community and national contexts. This will allow an expansion of the workforce contributing to the TB cascade of care. To accomplish this expansion, key pharmacy supports such as educators and policymakers should aim to increase pharmacy students’ knowledge about public health and possible future roles that both strengthen the traditional pharmaceutical care roles at the interpersonal level and the larger subnational and national systems that support TB elimination.

## Figures and Tables

**Figure 1 healthcare-12-01137-f001:**
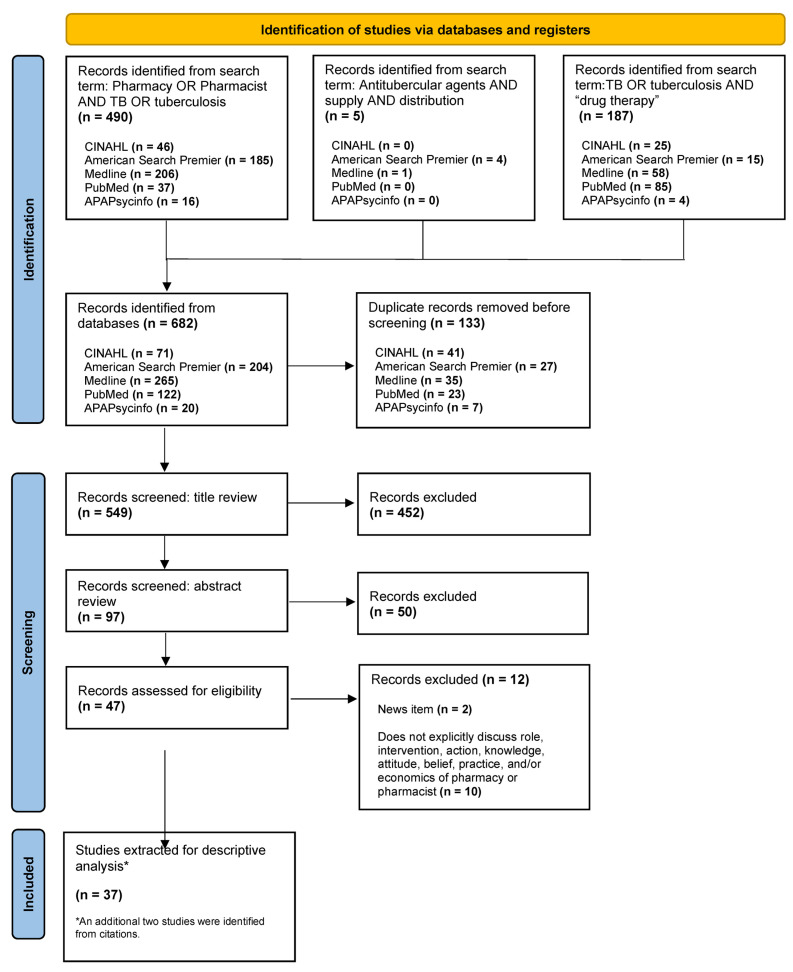
PRISMA flow diagram for this scoping review.

## Data Availability

Data are contained within the article.

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
