# Peer review of "Pharmacists’ Role in Global TB Elimination: Practices, Pitfalls, and Potential"

_healthcare, 2024, doi:10.3390/healthcare12111137_

Round 1
Reviewer 1 Report
Comments and Suggestions for Authors
The review provides a comprehensive description of the various roles pharmacists play in different stages of Tuberculosis (TB) management, from prevention to treatment. It summarizes articles from various databases and outlines nine detailed roles of pharmacists in TB management. The review is clear and thorough, with the introduction effectively using statistics to underscore the importance of TB and the significance of the review. However, there are a few minor points that could be addressed to improve readability.
1. In Table 2, there is a column where both "achieved" and "failed" are marked as "Yes". The meaning of this is unclear. The review does not provide a detailed description of the table in the text, and the table footnotes do not offer any explanation. It is confusing why a role can be both achieved and failed. Could the author clarify this?
2. Similarly, in Table 1, the World Bank country classifications are provided, but the abbreviations are not defined, and the author does not refer to these classifications in the text. A brief sentence of these classifications could be added to improve readability.
3. The length of the content is not evenly distributed. For instance, Role 1 is discussed very concisely within half a page, while Role 2 is discussed in great detail over 2.5 pages. Were these sections drafted by different authors? Could the author summarize more and condense the content, presenting key points for each role more clearly? This would make it easier for readers to follow.
Reviewer 2 Report
Comments and Suggestions for Authors
Manuscript ID: healthcare-2967695
Title: “Pharmacists’ role in global TB elimination: Practices, pitfalls, and potential” by Alina Cernasev, Jonathan Stillo, Jolie Black, Mythili Batchu, Elaina Bell, and Cynthia A Tschampl.
The role of pharmacists in TB care was described and supported in a joint statement by the World Health Organization (WHO) and the International Pharmaceutical Federation in 2011. This review is based on 37 articles from initial 682 hits on the subject concerning pharmacist contribution to global TB elimination. The authors found nine roles of pharmacists in TB prevention concluding that more pharmacists are already expanding their roles, while education is good to broaden this initiative.
Major comment:
Although interesting, there are notable shortcomings in this review that need addressing. Firstly, the length of the text is excessive for a review, and its article-like format may not be the most suitable. The sections under ‘Results’ are sentences from some of the 37 articles without any attempt to summarize them with conclusion, to support the individual sections. Integrating and expanding the 'Discussion' section could enhance coherence. Additionally, incorporating tables and figures to condense information and illustrate identified roles would be beneficial.
A separate section dedicated to current challenges faced by pharmacists in tuberculosis (TB) management should be included and thoroughly discussed. Previous research has highlighted deficiencies in how pharmacy professionals handle individuals with presumptive TB, including inappropriate over-the-counter sales of medicines like fluoroquinolone antibiotics, steroids, and cough suppressants, which can mask TB symptoms (see doi:10.1016/j.lansea.2023.100152, doi:10.1016/S1473-3099(16)30215-8).
Furthermore, the review disproportionately diminishes the roles of other professionals crucial to TB elimination, such as clinicians and TB diagnostic labs. These professionals play significant roles in diagnostics (e.g., sputum, x-ray, IGRA), drug prescription, and patient follow-up and monitoring. TB elimination is a collaborative effort among various professionals and should be acknowledged as such.
To enhance comprehension, incorporating a flow chart depicting total, excluded, and included articles would be beneficial. Additionally, it's advisable to include a table listing the 37 selected articles along with their respective references.
Row 64: ‘Two billion people are infected with TB every year’. Do the authors mean newly infected? In that case 10 million according to WHO.
Row 64-64: ‘A recent WHO report emphasized an 64 increase of 3.6% in 2021 compared to 2020, which reversed the two-decade trend of a 2% 65 decrease annually’. An even more recent WHO report shows a decrease once again.
Row 695: ‘However, there are challenges to its adoption, such as ensuring the accessibility of rifapentine which’. A bit of the sentence is missing.
Round 2
Reviewer 2 Report
Comments and Suggestions for Authors
The authors have now thoroughly worked on the manuscript and adequately addressed my questions.